# Investigation of Antioxidant Synergisms and Antagonisms among Phenolic Acids in the Model Matrices Using FRAP and ORAC Methods

**DOI:** 10.3390/antiox11091784

**Published:** 2022-09-09

**Authors:** Danijela Skroza, Vida Šimat, Lucija Vrdoljak, Nina Jolić, Anica Skelin, Martina Čagalj, Roberta Frleta, Ivana Generalić Mekinić

**Affiliations:** 1Department of Food Technology and Biotechnology, Faculty of Chemistry and Technology, University of Split, R. Boškovića 35, HR-21000 Split, Croatia; 2University Department of Marine Studies, University of Split, R. Boškovića 37, HR-21000 Split, Croatia; 3Center of Excellence for Science and Technology-Integration of Mediterranean Region (STIM), Faculty of Science, University of Split, HR-21000 Split, Croatia

**Keywords:** phenolic acids, phenolic mixtures, interaction effect, antioxidant activity, FRAP, ORAC

## Abstract

The total antioxidant potential of a sample cannot be predicted from the antioxidant activity of its compounds; thus, scientists usually explain the overall activity through their combined effects (synergistic, antagonistic, or additive). Phenolic compounds are one of the most powerful and widely investigated antioxidants, but there is a lack of information about their molecular interactions. This study aimed to investigate the individual and combined antioxidant activity of equimolar mixtures (binary, ternary, quaternary, and quinary) of 10 phenolic acids (protocatechuic, gentisic, gallic, vanillic, syringic, *p*-coumaric, caffeic, ferulic, sinapic, and rosmarinic acid) at different concentrations using ferric reducing antioxidant power (FRAP) and oxygen radical absorbance capacity (ORAC) assays. Gallic acid showed the highest antioxidant activity, determined using the FRAP assay (494–5033 µM Fe^2+^) and rosmarinic acid with the ORAC assay (50–92 µM Trolox Equivalents (TE)), while the lowest antioxidant potential was observed for *p*-coumaric acid (FRAP 24–113 µM Fe^2+^ and ORAC 20–33 µM TE). The synergistic effect (by FRAP) in the equimolar mixtures of hydroxybenzoic acids was confirmed for a large number of tested mixtures, especially at low concentrations. All mixtures containing gentisic acid showed a synergistic effect (28–89% difference). Using the ORAC method, only two mixtures of hydroxybenzoic acids showed an antagonistic effect, namely a mixture of gentisic + syringic acids (−24% difference) and gallic + vanillic acids (−30% difference), while all other mixtures showed a synergistic effect in a range of 26–236% difference. Among mixtures of hydroxycinnamic acids, the highest synergistic effect was observed for the mixtures of *p*-coumaric + ferulic acids and caffeic + sinapic acids with differences of 311% and 211%, respectively. The overall antioxidant activity of phenolic acids could be explained by the number or position of hydroxyl and/or methoxy functional groups as well as the compound concentration, but the influence of other parameters such as dissociation, intramolecular hydrogen bonds, and electron donating or withdrawing effect should not be neglected.

## 1. Introduction

The research effort concerning the antioxidant behavior of phenolic compounds has significantly increased in recent decades, but the knowledge about their interaction in model mixtures is still scarce. Among the diverse and complex groups of phenolics that include simple phenols, flavonoids, stilbenes, tannins, and others, phenolic acids are the most distributed in nature. They have been found in various plants, fruits, vegetables, beverages, and agro-food by-products where they contribute to organoleptic attributes such as color, flavor, and odor but their true merits are numerous positive biological activities such as antibacterial, anti-inflammatory, antiallergenic, anticancer, cytotoxic, antitumor, cardioprotective, and antioxidant, which is among the most investigated [1,2,3,4,5,6,7,8,9,10].

Phenolic acids are represented by two main classes: hydroxybenzoic and hydroxycinnamic acids, containing seven (C1–C6) and nine (C3–C6) carbon atoms, respectively. Each phenolic acid is composed of an aromatic ring with hydroxyl (–OH) and carboxyl (–COOH) groups, and the main difference in the structure of these groups is the presence of one additional double bond between the –COOH group and the aromatic ring [3,11,12,13]. The phenolic acids also differ in type, number, and position of the attached functional groups on the aromatic ring (–OH, methoxy (–OCH_3_)), and the research on their distribution is commonly used to find a relationship between structural features and compound activity, known as quantitative structure–activity relationship (QSAR) [14]. However, the knowledge of the mechanisms by which these molecules and their parts act in different reactions is limited. Scientific research indicated several factors with a possible impact on the mechanisms behind the compound’s activity. Among them, the number and position of hydroxyl groups and their methylation, the distance between phenyl and carboxylic groups, and the concentration of the compound are suggested [15,16,17,18,19,20,21].

Like other phenolics, phenolic acids demonstrate different mechanisms of antioxidant action such as reduction of agents by hydrogen donation, quenching of singlet oxygen, or acting as chelators and trappers of free radicals, so usually, methods used to analyze their antioxidant activity are based on different mechanisms [3,14,22,23]. These methods may be generally classified as electron transfer (ET) and hydrogen atom transfer (HAT)-based assays [24]. The most accepted and widely used assays for the determination of antioxidant activity are Folin–Ciocalteu, FRAP (ferric reducing antioxidant power), ABTS/TEAC (2,2′-azinobis (3-ethylbenzothiazoline 6-sulfonate radical scavenging activity/Trolox equivalent antioxidant capacity), DPPH (2,2-diphenyl-1-picrylhydrazyl radical scavenging activity), and ORAC (oxygen radical absorbance capacity) [23,24,25,26]. Among these, Folin–Ciocalteu, FRAP, ABTS/TEAC, CUPRAC (cupric reducing antioxidant capacity), and DPPH methods are ET-based assays that provide information about reducing the capacity of an antioxidant, while the ORAC method is based on the HAT reaction mechanism. The FRAP method is often used to measure the reducing power of different samples and is considered one of the fastest, simplest, and less expensive methods, with reproducible results in a wide range of concentrations. On the other hand, the ORAC method uses a biologically relevant radical source (peroxyl radical), thus, the obtained activity could be used for interpreting activity in various biological systems [23,24,27]. For these reasons, the results obtained using different methods must be interpreted carefully, as due to differences in their mechanisms, the correlations between the obtained result often fail [23,26,28].

Although the antioxidant activity of phenolic acids is well studied using both in vitro and in vivo methods, the mechanisms of their action remain unclear and/or undefined [17,18,19]. An important factor that should be considered is their mutual interactions which can be synergistic, antagonistic, or additive (no interaction). Several studies aimed to investigate these interactions among phenolic acids using different antioxidant assays and confirmed both the occurrence of synergistic as well as antagonistic interactions [13,14,26,27,29,30,31,32,33,34]. The authors emphasized the influence of chemical structure and used concentrations on the overall activity of the tested mixtures. The efficiency of these interactions is also widely used to explain the activities of phenolic-rich extracts, where the dominant components cannot be identified as carriers of the total antioxidant activity [10,31,35].

In this regard, this study aimed to evaluate the antioxidant potential (reducing and free scavenging activity) of individual phenolic acids (protocatechuic, gentisic, gallic, vanillic, syringic, *p*-coumaric, caffeic, ferulic, sinapic, and rosmarinic) and their interactions in binary, ternary, quaternary, and quinary equimolar mixtures at different concentrations using FRAP and ORAC assays.

## 2. Materials and Methods

### 2.1. Preparation of Standard Solutions and Model Mixture

All used reagents and solvents were analytical or higher grade and purchased from Sigma (Sigma-Aldrich GmbH, Steinheim, Germany), Alkaloid AD (Skopje, North Macedonia), Merck (Darmstadt, Germany), Fluka (Buch, Switzerland), and Kemika (Zagreb, Croatia). The solutions of hydroxybenzoic acids (protocatechuic, gentisic, gallic, vanillic, and syringic) and hydroxycinnamic acids (*p*-coumaric, caffeic, ferulic, sinapic, and rosmarinic) (Sigma-Aldrich GmbH, Steinheim, Germany) shown in Table 1 were dissolved in an ethanol/water mixture (80:20, by volume) to the final concentration of 1000 µM. The experiment was divided in two parts. Firstly, all phenolic acids were individually tested for antioxidant activity at the concentrations of 2.5 and 5 µM in the ORAC assay, and 100, 500, and 1000 µM in the FRAP assay. Thereafter, the phenolic acids were mixed in binary, ternary, quaternary, and quinary equimolar combinations to reach the concentrations of 5 µM for ORAC and 100, 500, and 1000 µM for the FRAP assay.

### 2.2. Evaluation of the Antioxidant Activity

#### 2.2.1. FRAP (Ferric Reducing Antioxidant Power) Assay

The reducing power of the samples detected with the FRAP method was measured according to the procedure described by Skroza et al. [36] and measurements were performed on a Tecan MicroPlate Reader, model Sunrise (Tecan Group Ltd., Männedorf, Switzerland). Analyses were completed in triplicates and results are expressed as µM of Fe^2+^.

#### 2.2.2. ORAC (Oxygen Radical Absorbance Capacity) Assay

Fluorimetric measurements in the ORAC assay were recorded on a microplate reader (Synergy HTX Multi-Mode Reader, BioTek Instruments, Inc., Winooski, VT, USA) following the procedure of Čagalj et al. [37]. The reaction was observed for 80 min and the results of three replicates are expressed in µM of Trolox Equivalents (µM TE).

### 2.3. Interaction and Statistical Analysis

The obtained results were analyzed using GraphPad Prism Version 4.03 for Windows (GraphPad Software, San Diego, CA, USA).

The interactions between phenolic acids were described as the difference in antioxidant activity between experimental and theoretical (calculated) values using the equation (Equation (1)) [36,38]:*Difference* (%) = ((*Combination ab* × 100)/(*Individual a* + *Individual b*)) − 100(1)
where combination *ab* is an experimentally obtained result for the binary mixture, while each *a*/*b* value was calculated individually for each compound. The theoretical values for each compound were calculated by dividing the experimental values by the number of compounds in the mixtures. Likewise, for ternary, quaternary, and quinary mixtures, the difference was calculated by subtracting the average of the individual three, four, or five compounds from the combination (Equations (2)–(4)):*Difference* (%) = ((*Combination abc* × 100)/(*a* + *b* + *c*)) − 100(2)
*Difference* (%) = ((*Combination abcd* × 100)/(*a* + *b* + *c* + *d*)) − 100(3)
*Difference* (%) = ((*Combination abcde* × 100)/(*a* + *b* + *c* + *d* + *e*)) − 100(4)

The obtained interactions (% difference) were used to determine potential synergistic (positive values, difference (%) > 0) or antagonistic (negative values, difference (%) < 0) effects. The additive effect was considered for the difference (%) ≅ 0 ± 5% when it can be considered that there was no interaction.

## 3. Results and Discussion

### 3.1. Antioxidant Activity of Individual Phenolic Acids

Previous studies confirmed that the number and arrangement of –OH and –OCH_3_ groups (their mutual position) affect the antioxidant activity of the phenolic acids [14,27,31,39], but also other parameters, such as ionization, dissociation, and the rate constants of radical scavenging, resonance, solvent solvation effects, intramolecular hydrogen bonds, bond dissociation enthalpy, etc., should also be taken in consideration as reported by Sroka [16], Hanscha at al. [15], Foti et al. [17] Lucarini and Pedulli [18], Hang et al. [21], and Biela et al. [20].

The antioxidant activities of individual phenolic acids at different concentrations are provided in Table 2. Among the hydroxybenzoic phenolic acids, gallic acid showed the highest FRAP value, while the lowest activity was detected for vanillic acid. This was observed for all tested concentrations and linear dependence was confirmed, as expected. The highest reducing power of gallic acid could be related to its chemical structure and three –OH groups located at positions 3, 4, and 5. The gentisic acid, with two –OH groups in *para*-position to each other (at positions 2 and 5), and protocatechuic acid with two –OH groups at positions 3 and 4 (catechol structure) showed similar FRAP values at low concentrations, while at higher concentrations gentisic acid was superior. Considering the molecular structure, the activity of gentisic acid, an active metabolite of salicylic acid degradation, was investigated by Mardani-Ghahfarokhi and Farhoosh [40], and the authors reported its higher activity in comparison to α-resorcylic acid with the same type and number of substituents, but in *meta*-position. Sroka and Cisowski [16] reported that acids with two –OH groups exhibited higher antioxidant activity than those possessing only one. Cuvelier et al. [41] concluded that the introduction of –OH in *para*- or *ortho*-position enhances compound activity which increases also with the –OCH_3_ substitution in *ortho*-positions to –OH group. The –COOH group, as one of the most common electron-withdrawing groups (containing an atom with a positive charge directly attached to a benzene ring) with a relatively strong effect, can affect other substituents and it has been known that this effect is the strongest on –OH groups in the *para*-position, and weaker on those in the *meta*-position [20,42]. Biela et al. [20] reported that phenolic acids in aqueous solutions are mostly dissociated (fully deprotonated) to carboxylate anions so the key role in the activity is carried by their phenolic –OH groups. It has been reported that the carboxylic substituent is a weak donor in the *meta*-position and has no effect in the *para*-position in comparison to the –COOH. Furthermore, the presence of other substituents, such as carboxylate anions, –OH and –OCH_3_ can also form intramolecular hydrogen bonds that can affect the activity. The authors concluded also that –OH and –OCH_3_ show an electron-donating effect in the *para*-position and an electron-withdrawing effect in the *meta*-position. Syringic acid, with two –OCH_3_ groups at positions 3 and 5 (*meta*-position), and one –OH at the *ortho*-position, showed an increase in activity at higher concentrations reaching 3207 µM Fe^2+^ at 1000 µM, while vanillic acid, with an –OH group in the *para*-position, and –OCH_3_ group in the *meta*-position had the lowest FRAP value. Spiegel et al. [27] investigated 22 phenolic acids using an FRAP assay and reported that the main structural feature of good antioxidants are two or more –OH groups in *ortho*- and *para*-positions, but the importance of the inductive effect of the carboxylic group should not be neglected. Protocatechuic acid has two –OH groups, the same as gentisic acid, but it showed a lower reducing effect so it can be concluded that again, besides the number and arrangement of attached functional groups (in this case, *ortho*- or *para*-position), the intramolecular hydrogen bonds between two –OH groups also have a notable impact. Additionally, intramolecular hydrogen bonds between the –OH and –COOH groups may affect antioxidant activity. The obtained results showed that hydroxylation at positions 2 and 5 and one intramolecular hydrogen bond between them contributes to the reducing ability of the compound [17,18,27]. Rice-Evans et al. [43] reported that the insertion of an additional –OH group at position 2 of hydroxybenzoic acids decreases the overall antioxidant capacity, while Sroka and Cisowski [16] showed that the antioxidant activity of phenolic acids correlates with the number of –OH groups in *ortho*- and *para*-positions, but also reported the importance of the position of –COOH and acetyl group near the –OH groups. Foti et al. [17], except for position and number of substituents, also indicate the importance of resonance stabilization and intramolecular hydrogen bonds between them, while Biskup [11] reported that the functional group binding site and the type of substitute affect the activity. Spiegel et al. [27] observed that the position of the second –OH group affected the reducing capacity and that two or more –OH groups located either in the vicinal position or in the opposite position to each other resulted in higher antioxidant activity. They also explained differences in antioxidant activity of phenolic acids using resonance stabilization of radicals by intermolecular hydrogen bonds between functional groups and a polar solvent. The influence of the hydrogen bonds is also discussed by Foti et al. [17] where the authors reported that only compounds that are non-hydrogen-bonded (free) possess activity (electron transfer mechanism) and that the rate of reaction depends on the strength of the hydrogen bond as well as on the used solvent (methanol or ethanol).

The FRAP values of the hydroxycinnamic acids also showed a linear correlation with concentration (Table 2). Similarly, FRAP values increased with the introduction of –OH and –OCH_3_ groups, which is in correlation with previous reports [14,27]. Among hydroxycinnamic acids, caffeic and rosmarinic acids had the highest FRAP values, while *p*-coumaric acid, with one –OH group, showed the lowest FRAP at all tested concentrations. Exceptionally good reducing power was also observed for sinapic and ferulic acids. In addition to one –OH group, these two acids also have –OCH_3_ groups, sinapic acid two and ferulic only one. The rosmarinic acid, an ester of caffeic acid and 3,4-dihydroxyphenyllactic acid (diphenolic compound), also stands out, with FRAP values 2-fold higher than those obtained by caffeic acid alone at 1000 µM. This can be also connected with its structure (two phenolic rings with two –OH groups in the *ortho*-position and an unsaturated double bond and –COOH between them). Cao et al. [44] investigated the antioxidant activity of this compound given its molecular structure and reported a stronger electron-donating capability of the B ring. According to the authors, its activity is a result of the H-abstraction reactions on both rings. Therefore, the activity of rosmarinic acid can be related to two phenolic groups in both rings. The good reducing power of caffeic acid can be related to the catechol structure and distance between the –COOH group and functional groups [14,16,17,27]. It has also been reported that hydroxybenzoic acids have lower antioxidant activity in comparison to hydroxycinnamic acids when they have the same substituents at positions 2–6 [14,20,27,41,43,45]. These observations are also confirmed in the present study where syringic acid showed higher reducing activity than sinapic acid, and ferulic acid was superior to vanillic acid. In the case of the ORAC method, the values for ferulic and vanillic acids at 2.5 µM were similar, but at higher concentrations better activity was detected for vanillic acid.

The results of antioxidant activity tested using the ORAC method at two concentrations (2.5 and 5 µM) are shown in Table 2. Among hydroxybenzoic acids, protocatechuic acid showed the highest activity, while gallic acid had the lowest activity. At higher concentrations, 21–55% higher activity was detected for all acids, with exception of the gallic acid which had 19% higher activity. Protocatechuic acid, with two –OH groups at positions 2 and 3, had higher activity than other acids with the same number of –OH groups at other positions (gentisic acid) or even –OCH_3_ groups (one in vanillic acid and two in syringic acid structure). Sroka and Cisowski [16] in their study also reported higher activity of protocatechuic acid against DPPH free radicals in comparison to the gentisic acid. Hang et al. [21] reported the influence of –OH and/or –OCH_3_ groups at position 3 and/or 5 on the hydrogen transfer mechanism (characterized by bond-dissociation energy) using the hydroperoxyl radical scavenging assay. They conclude that the intramolecular hydrogen bond between the –OH group at the *ortho*-position and the –COOH group could be the main reason for the highest reduction of bond-dissociation energy, which indicated weaker antioxidant activity (radical scavenging activity followed the order: syringic acid > gentisic acid > gallic acid > vanillic acid > protocatechuic acid). For higher antioxidant activity measured using ORAC, the location of –OH groups in vicinal or in the opposite position side of the ring seems to be a more important factor than only the number of –OH groups.

The ORAC test for hydroxycinnamic acids (Table 2) showed the lowest results for *p*-coumaric acid, while rosmarinic acid had the highest antioxidant effect at tested concentrations. This could be again related to its chemical structure and four –OH groups. *p*-coumaric acid was previously described as a weak antioxidant [29], what is also confirmed in this research where the obtained results showed that caffeic, ferulic, and sinapic acids had 2-fold higher activity than this compound. While *p*-coumaric acid has only one –OH group, other acids have additional –OH and/or –OCH_3_ groups. At the concentration of 5 µM, all acids showed approximately 45% higher values. Ferulic (one –OH and one –OCH_3_) and sinapic (one –OH and two –OCH_3_) acids showed almost the same activity, indicating that the additional –OCH_3_ group does not have a significant impact on the activity. Interestingly, caffeic acid with a catechol group at the same position as protocatechuic acid showed similar antioxidant activity as this hydroxybenzoic acid, in accordance with the results of Sroka and Cisowski [16] who also confirmed a similar free radical scavenging activity. The authors pointed out that although these compounds have a similar model of substitution of the –OH and the incorporation of –CH_2_ between –COOH and the phenyl group does not increase the antiradical activity of 3,- and 4- substituted acids. This is also in accordance with the conclusions that the catechol group enhances the radical scavenging activity of the compound [26,27,29]. However, ferulic acid was not effective in scavenging peroxyl radicals such as vanillic acid with the same structural features (–OH and –OCH_3_ group), not in accordance with previous reports [26,27,29]. Additionally, greater distances between –COOH groups from the methoxylated ring do not enhance the antioxidant effect, as previous studies by Mathew et al. [19] and Spiegel et al. [27] suggested. Lucarini and Pedulli [18] reported the importance of the reaction medium. In their study on free radical scavenging activity of peroxyl radicals in autooxidation reactions, they reported the connection of bond dissociation enthalpies and rate constants with the antioxidant compound structure. The authors would like to point out once more the lack of systematic research on the antioxidant activity of phenolic acids using the ORAC method and investigations of their structure–activity relations.

### 3.2. Antioxidant Activity of Equimolar Mixtures of Phenolic Acids

The results of the antioxidant activity of the equimolar mixtures of two or more phenolic acids tested at different concentrations are shown in Table 3, Table 4, Table 5 and Table 6. Based on the obtained data, the potential interaction (synergistic/additive/antagonistic) was determined and expressed as a percentage of the difference (%) between the experimental and theoretical (calculated) FRAP and ORAC values.

In the binary mixtures of hydroxybenzoic acids, at 100 µM, all mixtures containing gentisic acid showed a synergistic effect (28–89% difference). The mixture of protocatechuic and syringic acid showed an additive effect, while all others showed an antagonistic effect (up to −58% difference). At a concentration of 500 µM the synergic effect was observed only for the mixture of gentisic + syringic acids, while at 1000 µM the synergistic effect was not confirmed. Among ternary mixtures, it is interesting to highlight the mixture protocatechuic + gentisic + syringic acids, which showed the highest reducing power and the greatest synergistic effect (174% difference) at the lowest tested concentration. In comparison to the mixture of protocatechuic + gentisic acids and gentisic + syringic acids that also showed a high synergistic effect, and the mixture of these three phenolic acids showed a higher overall reducing capacity. When protocatechuic, gallic, and vanillic acids were combined they retained the antagonistic effect observed for mixtures of protocatechuic + gallic acid acids and protocatechuic + vanillic acids at all tested concentrations. Again, the mixtures containing either protocatechuic, gallic, and/or syringic acid along with gentisic acid show a synergistic effect leading to a conclusion that gentisic acid is a key component for the synergistic effect of these mixtures. However, the synergistic effect for protocatechuic + gentisic + vanillic acids and protocatechuic + gentisic + syringic acids observed at the concentration of 100 µM was confirmed also at 500 µM but not at a higher concentration where it was only detected for the protocatechuic + gentisic + gallic acids mixture. These results indicated that along with the compound ratio in the mixture, the compound concentration is also an important factor. The quaternary and quinary equimolar combinations showed a synergic effect only at the concentration of 100 µM, with the exception of the mixture of gallic + vanillic + syringic + protocatechuic acids which showed an antagonistic effect at all tested concentrations.

The results of the reducing capacity for the mixtures of hydroxycinnamic acids are shown in Table 4. In contrast to the results for interactions obtained for hydroxybenzoic acids, a large number of tested mixtures of cinnamic acids showed lower antioxidant activity compared to the expected theoretical values, which indicated an antagonistic effect between these compounds. Among binary combinations, ferulic + sinapic acids showed the highest reducing power at 100 µM (501 µM Fe^2+^), while at other concentrations the mixture of *p*-coumaric + ferulic acids exhibited the lowest activity. At the concentration of 100 µM, the greatest positive difference between theoretical and expected FRAP values, indicating the higher synergistic effect, was observed for the following mixtures: *p*-coumaric + sinapic acids (72% difference) and ferulic + sinapic acids (102% difference), while other mixtures showed an antagonistic effect. With the increase in concentration, the number of mixtures showing a synergistic effect rose to five at 500 µM and nine at 1000 µM. At the first higher concentration, only the combination of ferulic + rosmarinic acids had an antagonistic effect while at 1000 µM they showed only an additive interaction. The addition of *p*-coumaric acid to the mixture of ferulic and rosmarinic acid, at both concentrations of 500 and 1000 µM, resulted in a synergistic effect of the mixture (127 and 48% difference, respectively) which was strange due to the low FRAP value of this compound and only one –OH group in its structure. In ternary mixtures, only *p*-coumaric + caffeic + rosmarinic acids showed a synergistic effect at all tested concentrations, but the decrease in the difference with the increase in the concentration was recorded (127% at 100 µM > 48% at 500 µM > 11% at 1000 µM). The mixture of caffeic + sinapic + rosmarinic acids, with the highest number of –OH (seven) and –OCH_3_ (two) groups showed an antagonistic effect at 100 µM and a synergistic effect at 500 and 1000 µM. In mixtures of *p*-coumaric + caffeic + sinapic acids and caffeic + sinapic + rosmarinic acids, an antagonistic effect passed to synergistic at 1000 µM. Among quaternary and quinary mixtures, at 1000 µM only mixtures of caffeic + ferulic + sinapic + rosmarinic acids, mixtures of caffeic + sinapic + rosmarinic + *p*-coumaric acids and *p*-coumaric + caffeic + ferulic + sinapic + rosmarinic acids showed a weak synergistic effect.

Hajimehdipoor et al. [32] confirmed the synergistic effect of the caffeic + rosmarinic acids mixture (38% difference) in different binary combinations. The authors mixed binary combinations of hydroxybenzoic and hydroxycinnamic acids (gallic, rosmarinic, caffeic, and chlorogenic), alone and with flavonoids (quercetin and rutine) and reported that binary mixtures show stronger synergistic effects than their ternary combinations. Olszowy-Tomczyk [23] also reviewed the available information in the literature about interactions among compounds in the binary mixtures of phenolic acids with other phenolics (flavonoids, catechins, stilbenes, etc.). Differences between the experimental and theoretical values for antioxidant activity among phenolic acids were observed also in binary mixtures of gallic + protocatechuic acids, gallic + vanillic acids [31], rosmarinic + caffeic acids [29], and gallic + caffeic acids [23].

The results of the theoretical and experimental ORAC values of phenolic mixtures, as well as their interactions, are presented in Table 5 and Table 6. Only two mixtures of hydroxybenzoic acids showed an antagonistic effect, namely a mixture of gentisic + syringic acids (−24% difference) and gallic + vanillic acids (−30% difference), while all other mixtures had a synergistic effect. The antioxidant activities and synergistic effects of binary mixtures were higher in comparison to mixtures of three or more acids. The highest synergistic effect was observed for the mixture of gentisic + syringic acids with a total of two –OCH_3_ and three –OH groups while the mixture of gallic + syringic acids with the highest number of substituents, four –OH and two –OCH_3_ groups, showed an antagonistic effect. In the ORAC method, the presence of gentisic acid and protocatechuic acid resulted in a synergistic effect of the mixtures that contain these substances which could indicate that the presence of two –OH groups on the benzene ring (in *ortho*- or *para*-positions) is most likely responsible for this effect. In ternary, quaternary, and quinary mixtures, all combinations showed a synergistic effect except the mixture of protocatechuic + gallic + syringic acids which showed an additive effect. When gentisic acid was added to this mixture the effect was again synergistic.

The ORAC results obtained for the mixtures of hydroxycinnamic acids were lower and only a few mixtures showed the synergistic effect. The best antioxidant potential was confirmed for the mixture of *p*-coumaric + ferulic acids (160 µM TE) with a difference of 311% and a mixture of caffeic + sinapic acids (162 µM TE) with a difference of 211%. A positive interaction was observed also for the mixture of ferulic + rosmarinic acids (127%) and caffeic + rosmarinic acids (115%). Peyrat-Maillard et al. [29] also confirmed synergistic interaction between caffeic and rosmarinic acid at concentrations up to 5 µM by ORAC assay but in their study the concentration showed no effect on the interaction. In ternary, quaternary, and quinary mixtures of hydroxycinnamic acids only mixtures of *p*-coumaric + caffeic + ferulic acids, *p*-coumaric + caffeic + rosmarinic acids, and rosmarinic + *p*-coumaric + ferulic + sinapic acids showed a synergistic effect, while others showed a slight antagonistic or additive effect. In contrast to hydroxybenzoic acids, the number of the −OCH_3_ group in the structure of hydroxycinnamic acids (e.g., in ternary, quaternary, and quinary mixtures with ferulic or sinapic acids) cannot be related to their higher antioxidant activity.

Palafox Carlos et al. [31] suggested that gallic, protocatechuic, and vanillic acids interact in a synergic way. Using a DPPH assay, the authors also confirmed the synergistic effect of the gallic and protocatechuic acid mixture relating this effect to the chemical structure of the compounds and the presence of the hydroxyl group. On the other hand, in their study, the mixture of protocatechuic and vanillic acid showed an antagonistic effect. Some authors suggested that interactions are concentration-related, rather than structure-related [39] or that presence or absence of the catechol group in the chemical structure of the compounds from the mixtures contributes to their synergic effect [13,46]. They investigated the interaction effect between caffeic, ferulic, and rosmarinic acid at different concentrations (50, 100, 200, and 250 µM) using the Briggs–Rauscher assay and reported the synergistic effect of the mixtures at concentrations ranging from 50 to 200 µM, and strong antagonism at 250 µM. The authors concluded that the antioxidant activity depends on compound structure (number and distribution of substituents) and concentration, which was opposite to some of the results obtained in the present study.

## 4. Conclusions

The results indicated that differences in antioxidant activity of the tested phenolic acids depend on their structure, as expected, regarding not only the type, number, and arrangement of substituents but also the compound concentration. The additional number of the –OCH_3_ groups in the same positions in the phenolic ring in the hydroxybenzoic acids resulted in higher activity in comparison to the hydroxycinnamic acids with the same structural features. Among individual hydroxybenzoic acids, gallic acid showed the highest reducing activity, while the lowest activity was recorded with the ORAC assay. Among hydroxycinnamic acids the *p*-coumaric acid showed the lowest activity, using both methods at all tested concentrations. In the mixtures, synergistic effects were detected in several combinations, but special attention should be devoted to hydroxybenzoic acid mixtures containing gentisic acid, especially at lower concentrations where in all cases the positive differences were calculated. Similarly, in the ORAC method the presence of gentisic acid resulted in a synergistic effect of the mixtures, while low activity of the gallic acid obviously influenced the overall mixture activity as lower antioxidant or antagonistic effects are detected. Furthermore, it is obvious that other parameters such as the applied antioxidant method and solvent medium, the position of functional groups in relation to the –COOH group and other groups attached to the ring, ionization and bond dissociation enthalpies, intramolecular hydrogen bonding, etc., that were discussed but not investigated in this study, should be taken in consideration in further studies since they might have an impact on the overall antioxidant activity of the compounds and their mixtures.

## Figures and Tables

**Table 1 antioxidants-11-01784-t001:** List and structural features of the investigated phenolic acids.

	Common Name	IUPAC Name	R1	R2	R3
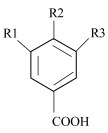	** *Hydroxybenzoic acids* **				
	Protocatechuic acid	3,4-dihydroxybenzoic acid	OH	OH	H
	Gentisic acid	2,5-dihydroxybenzoic acid	OH	H	OH
	Gallic acid	3,4,5-trihydroxybenzoic acid	OH	OH	OH
	Vanillic acid	4-hydroxy-3-methoxybenzoic acid	OCH_3_	OH	H
	Syringic acid	4-hydroxy-3,5-dimethoxybenzoic acid	OCH_3_	OH	OCH_3_
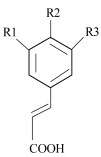	** *Hydroxycinnamic acids* **				
	*p*-coumaric acid	4-hydroxycinnamic acid	H	OH	H
	Caffeic acid	3,4-dihydroxycinnamic acid	OH	OH	H
	Ferulic acid	4-hydroxy-3-methoxycinnamic acid	OCH_3_	OH	H
	Sinapic acid	4-hydroxy-3,5-dimethoxycinnamic acid	OCH_3_	OH	OCH_3_
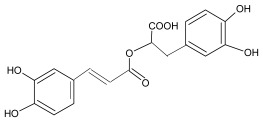	Rosmarinic acid	3,4-Dihydroxycinnamic acid (R)-1-carboxy-2-(3,4-dihydroxyphenyl)ethyl ester	Caffeic acid and 3,4-dihydroxyphenyllactic acid ester, with four OH groups		

**Table 2 antioxidants-11-01784-t002:** Antioxidant activity of individual phenolic acids at different concentrations using the FRAP and ORAC methods.

	FRAP (µM Fe^2+^)			ORAC (µM TE)	
	100 µM	500 µM	1000 µM	2.5 µM	5 µM
** *Hydroxybenzoic acids* **					
Protocatechuic acid	282 ± 5	1014 ± 14	2341 ± 32	30 ± 0.9	57 ± 2
Gentisic acid	294 ± 4	1710 ± 81	3186 ± 104	29 ± 1	53 ± 0
Gallic acid	494 ± 3	2478 ± 17	5033 ± 106	23 ± 1	28 ± 1
Vanillic acid	193 ± 2	542 ± 4	850 ± 13	26 ± 1	56 ± 1
Syringic acid	245 ± 5	1514 ± 19	3207 ± 111	29 ± 2	40 ± 1
** *Hydroxycinnamic acids* **					
*p*-coumaric acid	23.9 ± 2	61.8 ± 1	113 ± 3	20 ± 2	33 ± 1
Caffeic acid	476 ± 17	1321 ± 2	2125 ± 18	31 ± 2	60 ± 1
Ferulic acid	260 ± 6	885 ± 2	1706 ± 38	26 ± 2	45 ± 0
Sinapic acid	235 ± 8	1201 ± 0	2186 ± 24	25 ± 1	44 ± 1
Rosmarinic acid	413 ± 14	1803 ± 50	3656 ± 148	50 ± 2	92 ± 2

**Table 3 antioxidants-11-01784-t003:** Comparison of experimental and theoretical FRAP values and the interactions in equimolar phenolic mixtures of hydroxybenzoic acids at different concentrations.

	100 µM			500 µM			1000 µM		
	Experimental	Theoretical	Difference (%)	Experimental	Theoretical	Difference (%)	Experimental	Theoretical	Difference (%)
**Combination of two acids (1:1)**									
P + Ge	520 ± 2	288	**81**	1320 ± 15	1362	−3.1	2832 ± 22	2763	2.5
P + G	349 ± 7	388	−10	2181 ± 16	1746	**25**	3623 ± 7	3687	−1.7
P + V	101 ± 3	237	−58	559 ± 3	778	−28	1414 ± 8	1595	−11
P + Sy	274 ± 5	263	4.0	1280 ± 19	1264	**1.3**	2705 ± 10	2774	−2.5
Ge + G	542 ± 5	394	**38**	2034 ± 19	2094	−2.9	3937 ± 9	4109	−4.2
Ge + V	311 ± 3	243	**28**	825 ± 10	1126	−27	1440 ± 23	2018	−29
Ge + Sy	410 ± 3	270	**52**	2539 ± 4	1612	**58**	2974 ± 3	3197	−7.0
G + V	229 ± 5	343	−33	1237 ± 23	1510	−18	2861 ± 15	2942	−2.8
G + Sy	275 ± 2	370	−26	1814 ± 36	1996	−9.1	3599 ± 12	4120	−13
V + Sy	179 ± 0	219	−19	910 ± 5	1028	−12	1440 ± 23	2029	−29
**Combination of three acids (1:1:1)**									
P + Ge + G	166 ± 1	356	−54	2199 ± 28	1734	**27**	3811 ± 6	3520	**8.3**
P + Ge + V	317 ± 3	256	**24**	1338 ± 3	1089	**23**	1871 ± 5	2126	−12
P + Ge + Sy	749 ± 1	274	**174**	1509 ± 13	1413	**6.8**	2881 ± 6	2911	−1.1
P + G + V	248 ± 1	323	−23	1468 ± 15	1345	**9.2**	2591 ± 8	2741	−5.5
P + G + Sy	349 ± 6	340	2.6	1616 ± 8	1668	−3.2	2741 ± 50	3527	−22
P + V + Sy	210 ± 2	240	−13	947 ± 3	1023	−7.5	1786 ± 41	2133	−16
Ge + G + V	409 ± 3	327	**25**	1611.3 ± 8	1577	2.2	2430 ± 8	3023	−20
Ge + G + Sy	519 ± 3	344	**51**	1822.4 ± 12	1901	−4.1	3294 ± 47	3809	−14
G + V + Sy	245 ± 4	311	−21	1176.8 ± 6	1511	−22	2265 ± 21	3030	−25
V + Sy + Ge	335 ± 4	244	**37**	1023.5 ± 6	1255	−19	2014 ± 7	2414	−17
**Combination of four acids (1:1:1:1)**									
P + Ge + G + V	414 ± 3	316	**31**	1390 ± 2	1436	−3.2	2695 ± 20	2852	−5.5
P + Ge + V + Sy	367 ± 1	253	**45**	956 ± 4	1195	−20	1958 ± 4	2396	−18
Ge + G + V + Sy	412 ± 5	307	**34**	1236 ± 2	1561	−21	2470 ± 27	3069	−20
G + V + Sy + P	236 ± 0	303	−21	1291 ± 31	1387	−6.9	2789 ± 46	2858	−2.4
Sy + P + Ge + G	512 ± 5	329	**57**	1594 ± 15	1679	−5.1	3207 ± 43	3442	−6.8
**Combination of five acids (1:1:1:1:1)**									
P + Ge + G + V + Sy	439 ± 1	302	**46**	1257 ± 10	1452	−**13**	2876 ± 5	2923	−1.6

P—protocatechuic acid; Ge—gentisic acid; G—gallic acid; V—vanillic acid; Sy—syringic acid; *p*C—*p*-coumaric acid; C—caffeic acid; F—ferulic acid; Si—sinapic acid; R—rosmarinic acid; a difference (%) > 0 indicates a potential synergistic effect; a difference (%) < 0 shows an antagonistic and a difference (%) ≅ 0 or ± 5% shows an additive effect (no interaction).

**Table 4 antioxidants-11-01784-t004:** Comparison of experimental and theoretical FRAP values and the interactions in equimolar phenolic mixtures of hydroxycinnamic acids at different concentrations.

	100 µM			500 µM			1000 µM		
	Experimental	Theoretical	Difference (%)	Experimental	Theoretical	Difference (%)	Experimental	Theoretical	Difference (%)
**Combination of two acids (1:1)**									
pC + C	161 ± 10	250	−36	779 ± 18	691	**13**	1479 ± 42	1119	**32**
pC + F	95 ± 2	142	−33	537 ± 31	4474	−88	1046 ± 9	909	**15**
pC + Si	223 ± 10	130	**72**	655 ± 32	631	3.8	1321 ± 39	1150	**15**
pC + R	197 ± 10	219	−10	1357 ± 26	932	**46**	2635 ± 404	1885	**40**
C + F	314 ± 2	368	−15	1227 ± 28	5103	−76	2049 ± 28	1915	**7.0**
C + Si	342 ± 20	3560	−4.0	1362 ± 14	1261	**8.0**	2542 ± 87	2155	**18**
C + R	428 ± 44	445	−3.8	1751 ± 155	1562	**12**	3347 ± 122	2890	**16**
F + Si	501 ± 5	248	**102**	1234 ± 25	5043	−76	2303 ± 62	1946	**18**
F + R	322 ± 7	336	−4.2	1467 ± 19	5344	−73	2812 ± 57	2681	4.9
Si + R	307 ± 3	324	−5.2	1674 ± 56	1502	**11**	3394 ± 55	2921	**16**
**Combination of three acids (1:1:1)**									
pC + C + F	181 ± 2	253	−28	716 ± 18	3423	−79	1472 ± 26	1315	**12**
pC + C + Si	190 ± 9	245	−23	849 ± 19	861	−1.4	1559 ± 21	1475	**5.7**
pC + C + R	691 ± 39	305	**127**	1567 ± 32	1062	**48**	2178 ± 19	1965	**11**
pC + F + Si	171 ± 3	173	−0.9	737 ± 60	3383	−78	1436 ± 58	1335	**7.6**
pC + F + R	63 ± 7	232	−73	954 ± 8	3584	−73	1780 ± 54	1825	−2.5
pC + Si + R	216 ± 6	224	−3.6	947 ± 6	1022	−7.3	1925 ± 51	1985	−3.0
C + Si + R	328 ± 10	375	−13	1617 ± 22	1442	**12**	3214 ± 197	2656	**21**
C + F + Si	257 ± 5	324	−21	1044 ± 31	3803	−73	2241 ± 30	2005	**12**
C + F + R	337 ± 9	383	−12	1102 ± 23	4003	−73	2429 ± 54	2495	−2.7
F + Si + R	314 ± 3	303	3.9	1331 ± 71	3963	−66	2810 ± 43	2516	**12**
**Combination of four acids (1:1:1:1)**									
pC + C + F + Si	187 ± 8	249	−25	741 ± 22	2867	−74	1596 ± 44	1532	4.2
pC + C + F + R	224 ± 2	293	−24	967 ± 42	3018	−68	1917 ± 10	1900	0.9
C + F + Si + R	248 ± 11	346	−28	1164 ± 12	3303	−65	2576 ± 113	2418	**6.5**
C + Si + R + pC	245 ± 4	287	−15	1120 ± 12	1097	2.2	2176 ± 24	2020	**7.7**
R + pC + F + Si	201 ± 11	233	−14	1064 ± 21	2988	−64	1947 ± 36	1915	1.7
**Combination of five acids (1:1:1:1:1)**									
pC + C + F + Si + R	247 ± 1	282	−12	1107 ± 39	2655	−58	2077 ± 35	1957	**6.1**

P—protocatechuic acid; Ge—gentisic acid; G—gallic acid; V—vanillic acid; Sy—syringic acid; *p*C—*p*-coumaric acid; C—caffeic acid; F—ferulic acid; Si—sinapic acid; R—rosmarinic acid; a difference (%) > 0 indicates a potential synergistic effect; a difference (%) < 0 shows an antagonistic and a difference (%) ≅ 0 or ± 5% shows an additive effect (no interaction).

**Table 5 antioxidants-11-01784-t005:** Comparison of theoretical and experimental ORAC values and the interaction of equimolar phenolic mixtures (% difference) of hydroxybenzoic acids at a concentration of 5 µM.

	Experimental	Theoretical	Difference (%)
**Combination of two acids (1:1)**			
P + Ge	150 ± 0.4	55	**172**
P + G	55 ± 2	43	**28**
P + V	148 ± 1	56	**162**
P + Sy	51 ± 0.1	49	4.1
Ge + G	77 ± 5	41	**89**
Ge + V	150 ± 1	55	**174**
Ge + Sy	158 ± 2	47	**236**
G + V	29 ± 0.7	42	−30
G + Sy	26 ± 6	34	−24
V + Sy	149 ± 0.7	48	**210**
**Combination of three acids (1:1:1)**			
P + Ge + G	56 ± 0.5	46	**22**
P + Ge + V	82 ± 2	55	**48**
P + Ge + Sy	69 ± 2	50	**38**
P + G + V	61 ± 2	47	**29**
P + G + Sy	43 ± 1	42	1.9
P + V + Sy	65 ± 4	51	**27**
Ge + G + V	66 ± 4	46	**45**
Ge + G + Sy	50 ± 2	41	**22**
G + V + Sy	53 ± 1	41	**28**
V + Sy + Ge	65 ± 1	50	**29**
**Combination of four acids (1:1:1:1)**			
P + Ge + G + V	77 ± 2	49	**58**
P + Ge + V + Sy	86 ± 4	52	**67**
Ge + G + V + Sy	67 ± 2	45	**50**
G + V + Sy + P	65 ± 4	45	**44**
Sy + P + Ge + G	62 ± 9	45	**39**
**Combination of five acids (1:1:1:1:1)**			
P + Ge + G + V + Sy	60 ± 2	47	**27**

P—protocatechuic acid; Ge—gentisic acid; G—gallic acid; V—vanillic acid; Sy—syringic acid; *p*C—*p*-coumaric acid; C—caffeic acid; F—ferulic acid; Si—sinapic acid; R—rosmarinic acid; a difference (%) > 0 indicates a potential synergistic effect; a difference (%) < 0 shows an antagonistic and a difference (%) ≅ 0 or ± 5% shows an additive effect (no interaction).

**Table 6 antioxidants-11-01784-t006:** Comparison of theoretical and experimental ORAC values and the interaction of equimolar phenolic mixtures (% difference) of hydroxycinnamic acids at a concentration of 5 µM.

	Experimental	Theoretical	Difference (%)
**Combination of two acids (1:1)**			
*p*C + C	52 ± 2	46	**13**
*p*C + F	160 ± 4	39	**311**
*p*C + Si	52 ± 3	39	**34**
*p*C + R	61 ± 7	63	−2.1
C + F	59 ± 3	49	**21**
C + Si	162 ± 4	52	**211**
C + R	163 ± 9	76	**115**
F + Si	51 ± 4	45	**13**
F + R	156 ± 1	69	**127**
Si + R	74 ± 4	68	**9**
**Combination of three acids (1:1:1)**			
*p*C + C + F	53 ± 1	46	**16**
*p*C + C + Si	47 ± 3	46	2.1
*p*C + C + R	66 ± 3	62	**7.3**
*p*C + F + Si	39 ± 1	41	−4.2
*p*C + F + R	55 ± 2	57	−3.6
*p*C + Si + R	54 ± 1	56	−3.7
C + Si + R	56 ± 2	65	−14
C + F + Si	43 ± 3	50	−13
C + F + R	60 ± 2	66	−8.4
F + Si + R	47 ± 2	61	−22
**Combination of four acids (1:1:1:1)**			
*p*C + C + F + Si	45 ± 2	45	1.7
*p*C + C + F + R	56 ± 2	57	−2.9
C + F + Si + R	52 ± 1	60	−14
C + Si + R + *p*C	52 ± 1	57	−9.2
R + *p*C + F + Si	77 ± 4	54	**44**
**Combination of five acids (1:1:1:1:1)**			
*p*C + C + F + Si + R	56 ± 3	55	2.7

P—protocatechuic acid; Ge—gentisic acid; G—gallic acid; V—vanillic acid; Sy—syringic acid; *p*C—*p*-coumaric acid; C—caffeic acid; F—ferulic acid; Si—sinapic acid; R—rosmarinic acid; a difference (%) > 0 indicates a potential synergistic effect; a difference (%) < 0 shows an antagonistic and a difference (%) ≅ 0 or ± 5% shows an additive effect (no interaction).

## Data Availability

The data presented in this study are available in the article.

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
