# Peer review of "Investigation of Antioxidant Synergisms and Antagonisms among Phenolic Acids in the Model Matrices Using FRAP and ORAC Methods"

_antioxidants, 2022, doi:10.3390/antiox11091784_

Round 1
Reviewer 1 Report
Authors performed extensive and very interesting experimental work. However, the manuscript needs substantial revision. After careful revision and improvement of discussion, it may represent excellent contribution to the field of phenolic acids antioxidant activity.
In Abstract, TE abbreviation is not introduced (lines 24, 25).
In Abstract, the sentence “The overall antioxidant activity of mixtures could be explained by the number or position of hydroxyl and/or methoxy functional groups as well as the applied concentration.” indicates that authors intend to perform the analysis of the effect of aromatic ring substituents on the observed activity. However, this is almost absent. Thus, appropriate attention should be devoted to this aspect which can increase the scientific value of the paper.
In tables, rules of uncertainties presentation are neglected. Uncertainty should be presented with single significant digit and the value should be accordingly rounded. Uncertainties with 5 significant are total non-sense. The first value in Table 2 should be written in this form: 282 +/- 5. In Table 3, one table entry does not contain decimal dot. In text, found values should be also rounded accordingly, it is meaningless to present percentages using 5 significant digits (for example, line 239), when only 2 or three digits may be exact.
In discussion, authors have made insufficient and not fully correct attempt to explain found differences in antioxidant activities of individual acids (lines 156-167). Identified problems:
(i) OH and OCH3 groups show different effect in ortho/para position (electron-donating) and meta position (electron-withdrawing), see Hansch et al.: Chem. Rev. 91, 165−195, 1991, https://doi.org/10.1021/cr00002a004.
(ii) Authors do not mention the presence of intramolecular hydrogen bonds between neighboring OCH3 and OH, or two/three OH groups at adjacent carbon atoms. These affect antioxidant action of studied acids, see for example Foti et al.: J. Org. Chem. 69, 2309–2314, 2004, https://doi.org/10.1021/jo035758q, and Lucarini & Pedulli: Chem. Soc. Rev. 39, 2106–2119, 2010, https://doi.org/10.1039/B901838G.
(iii) In the case of benzoic acid derivatives, electron-withdrawing effect of COOH group affects mainly OH group in para-position (C4), while its effect on a group in meta position (OH group at C5 in gentisic acid) is considerably weaker. Besides, the two OH groups of gentisic acid are mutually in para position that positively affects its antioxidant activity.
(iv) Phenolic acids show different acidities. COOH group deprotonation changes their activity. Have authors kept constant pH of the solutions? Has ionic strength of solutions been kept constant, or it was given only by the degree of acids dissociation? Moreover, the degree of dissociation of acids in prepared mixtures can be different in comparison to single acid solution because stronger acid suppresses the dissociation of the weaker one. Authors should discuss this issue.
The sentence “While the –OH group in the meta- position is considered a characteristic of a good antioxidant, apparently the –OCH3 group at the same position does not have the same effect...” may not be correct, see above-mentioned issues. Moreover, in meta position, OH and OCH3 groups show identical electron-withdrawing effect, see Hansch et al.: Chem. Rev. 91, 165−195, 1991, https://doi.org/10.1021/cr00002a004.
Authors are encouraged to read recent report on phenolic acids by Biela et al.: Phytochemistry 200, 113254, 2022, https://doi.org/10.1016/j.phytochem.2022.113254, where the mutual effects of substituents at the aromatic ring, as well as intramolecular hydrogen bonds are discussed in detail. The paper is also focused on possible active role of OCH3 groups, because O-CH3 bond is the most labile in studied methoxy derivatives of phenolic acids. There is also very recent report on the hydroxybenzoic acids by Hoa et al.: Phytochemistry 201, 113281, 2022, https://doi.org/10.1016/j.phytochem.2022.113281 available. The two Phytochemistry papers can be useful for discussion/explanation of observed differences in activity of studied acids.
Authors have not explained, how the theoretical values summarized in tables were calculated. These could be also presented with their uncertainties. It can help to identify whether there are significant differences between measured values and “expected” values from the experiments performed for individual acids.
In relation to theoretical values used in FRAP experiments with 1000 micromolar concentration for the combinations of two acids, another question can be raised. Why did authors not used the sum of values for 500 micromolar concentration of individual acids? This could be also relevant comparison which may bring different results. For example, for Ge+V combination, results for 500 micromolar concentrations predict the value of 2253, while data related to 1000 micromolar concentrations offer only the value of 2018. On the contrary, for G+Sy, theoretical value would be only 3991 instead of 4120. Such approach could be also applied for 1:1 combinations in ORAC assays, since data for 2.5 and 5 micromolar concentrations are available.
Authors do not mention several relevant papers focused on phenolic acids, e.g., Mathew et al.: J. Food Sci. Technol. 52, 5790−5798, 2015, https://doi.org/10.1007/s13197-014-1704-0 and Sroka & Cisowski: Food Chem. Toxicol. 41, 753−758, 2003, https://doi.org/10.1016/S0278-6915(02)00329-0. Authors could compare/verify their results with these works.
Conclusions should be more detailed. In recent form, it only summarizes generally known facts. This section should stress the most important results.
Author Response
We are very grateful to Reviewer for all the efforts that has been made to improve our paper. All comments and suggestions have been studied; they are explained, accepted and incorporated in the revised version of the manuscript (modifications are marked using Track changes).

Reviewer 2 Report
Thank you for the opportunity to review the article “Investigation of antioxidant synergisms and antagonisms among phenolic acids in the model matrices using FRAP and ORAC methods”. The manuscript mainly investigated the individual and combined antioxidant activity of equimolar mixtures (binary, ternary, quaternary, and quinary) of ten phenolic acids (pro-tocatechuic, gentisic, gallic, vanillic, syringic, p-coumaric, caffeic, ferulic, sinapic and rosmarinic acid) at different concentrations by ferric reducing antioxidant power (FRAP) and oxygen radical absorbance capacity (ORAC) assays. The results indicated that differences in antioxidant activity of the tested phenolic acids depend on their structure and arrangement of substituents on the basic molecule as well as on their concentration. The objective is clear. The paper is well written, easy to follow and persuasive, methodology is sound. However, there are some comments and suggestions for the authors in order to improve the quality of their manuscript:
1. How to determine whether individual phenolic acids are synergistic or additive by calculating a percentage of the difference (%) between the experimental and theoretical (calculated) FRAP and ORAC values?
2. How do you illustrate the antioxidant activity of phenolic acids by the FRAP and ORAC methods and investigations of their structure-activity relations?
3. The results of the ORAC and FRAP test for hydroxycinnamic acids (Table 2) showed rosmarinic acid to have the highest antioxidant effect at tested concentrations. Why does rosmarinic acid have the highest antioxidant activity? Please analyze it in detail (structural characteristics, etc.) ?
4. This manuscript mainly investigated the antioxidant capacity of individual and combined phenolic acids, and predicted the effects of their structure and arrangement of substituents on the basic molecule. Are there further experiments to prove the effects of specific functional groups?
5. Line 247: What is the specific role of gentisic acid? What functional group came into play?
6. Please check and modify the format of the manuscript according to the journal requirements.
7. There are some punctuation errors in Table 3 and 4, please check and modify.
Author Response

(The authors gave the same response as above.)

Round 2
Reviewer 1 Report
Revised version can be published.
Two formal imperfections have been found:
Line 38: please, modify the text: electron donating or withdrawing effects
Lines 158, 159: Supposedly authors meant: dissociation (not dielectric) constant and the rate constants of radical scavenging? Please, check.
After their correction, further review is not necessary.
Author Response
The second revision has been made according to the Reviewer suggestions.
